# Toxin Dynamics among Populations of the Bioluminescent HAB Species *Pyrodinium bahamense* from the Indian River Lagoon, FL

**DOI:** 10.3390/md22070311

**Published:** 2024-07-04

**Authors:** Kathleen D. Cusick, Bofan Wei, Sydney Hall, Nicole Brown, Edith A. Widder, Gregory L. Boyer

**Affiliations:** 1Department of Biological Sciences, University of Maryland Baltimore County, Baltimore, MD 21250, USA; nbrown14@umbc.edu; 2Department of Chemistry, College of Environmental Science and Forestry, State University of New York, Syracuse, NY 13210, USA; bwei101@esf.edu (B.W.); shall8@esf.edu (S.H.); glboyer@esf.edu (G.L.B.); 3Ocean Research and Conservation Association, Vero Beach, FL 32960, USA; ewidder@teamorca.org

**Keywords:** saxitoxin, *Pyrodinium bahamense*, *sxtA4*, harmful algal bloom

## Abstract

Dinoflagellate species that form some of the most frequent toxic blooms are also bioluminescent, yet the two traits are rarely linked when studying bloom development and persistence. *P. bahamense* is a toxic, bioluminescent dinoflagellate that previously bloomed in Florida with no known record of saxitoxin (STX) production. Over the past 20 years, STX was identified in *P. bahamense* populations. The goal of this study was to examine toxin dynamics and associated molecular mechanisms in spatially and temporally distinct *P. bahamense* populations from the Indian River Lagoon, FL. *SxtA4* is a key gene required for toxin biosynthesis. *SxtA4* genotype analysis was performed on individual cells from multiple sites. Cell abundance, toxin quota cell^−1^, and *sxtA4* and RubisCo (*rbcL*) transcript abundance were also measured. There was a significant negative correlation between cell abundance and toxin quota cell^−1^. While the *sxtA4+* genotype was dominant at all sites, its frequency varied, but it occurred at 90–100% in many samples. The underlying mechanism for toxin decrease with increased cell abundance remains unknown. However, a strong, statistically significant negative correlation was found between *stxA4* transcripts and the *sxtA4/rbcL* ratio, suggesting cells make fewer *sxtA4* transcripts as a bloom progresses. However, the influence of *sxtA4−* cells must also be considered. Future plans include bioluminescence measurements, normalized to a per cell basis, at sites when toxicity is measured along with concomitant quantification of *sxtA4* gene and transcript copy numbers as a means to elucidate whether changes in bloom toxicity are driven more at the genetic (emergence of *sxtA4−* cells) or transcriptional (repression of *sxtA4* in *sxtA4+* cells) level. Based on the results of this study, a model is proposed that links the combined traits of toxicity and bioluminescence in *P. bahamense* bloom development.

## 1. Introduction

Dinoflagellate species that form some of the most frequent toxic blooms are also bioluminescent, yet the two traits are rarely considered together when studying mechanisms underlying bloom development [1]. Overall, both traits occur in small percentages: of the ca. 1555 marine species, ~70 are bioluminescent, and ~100 produce toxins. Of the bioluminescent species, 12 are confirmed toxin producers. However, both traits, though primarily toxicity, can vary among strains of the same species. *Pyrodinium bahamense* is a toxic, bioluminescent dinoflagellate that occurs in both the Atlantic–Caribbean and the Indo-Pacific [2,3]. It produces the potent neurotoxin saxitoxin. It is the causative agent of the human illnesses paralytic shellfish poisoning (PSP) and saxitoxin pufferfish poisoning (SPFP) [4]. Toxic blooms in the Indo-Pacific are well-documented [5,6,7,8]. *P. bahamense* is the source of bioluminescence in the bioluminescent bays in Puerto Rico. These populations are not routinely measured for toxicity; in general, it is assumed they are non-toxic. However, cells/strains within the populations appear to harbor the genetic machinery for toxin synthesis (discussed below). Limited toxin analysis by our lab showed they produce little to no toxin: our analysis of a sample from July 2022 showed the average toxin quota per cell to be extremely low (0.3 pg). *P. bahamense* has bloomed in the Indian River Lagoon (IRL), along the east coast of Florida, for years with no known record of STX production. Blooms of *P. bahamense* are also the primary source of the summertime bioluminescence along the east coast of Florida. In the mid-2000s, STX was identified in *P. bahamense* populations in Florida [4,9], marking the first documented occurrence of toxin production in these populations. It is worth noting that *P. bahamense* blooms have been increasing in both duration and intensity, beginning earlier in the spring and extending until early winter (Florida Wildlife Commission HAB reports, Cusick pers. obs.). The role of toxicity, as well as its dynamics within these populations, has not been established.

The first gene in the STX biosynthetic pathway, *sxtA* [10], codes for a novel polyketide synthase composed of four catalytic domains (SxtA1, SxtA2, SxtA3, and SxtA4) [11]. Two different forms of SxtA occur as follows: one with all four domains (coded for by *sxtA1,2,3,4*) and one lacking the SxtA4 domain (and thus the *sxtA4* gene) [10]. *SxtA1*, coding for the N-terminal portion of SxtA, has been found in both toxic and non-toxic strains of toxic species as well as non-toxic species from other genera [12,13,14]. *SxtA4* [15] is specific for toxin synthesis, as biochemical and molecular data corroborate genetic screens indicating that *sxtA4* is essential for STX biosynthesis [10,12,15,16,17,18,19]. The Cusick lab previously developed a single-cell multiplex PCR assay targeting the *sxtA4* and the 18S rRNA genes for screening populations of *P. bahamense* [20] from various sites in the IRL and Mosquito Bay (MB), a bioluminescent bay in Puerto Rico. These results revealed that within *P*. *bahamense* populations, there exist both *sxtA4+* and *sxtA4−* cells, where *sxtA4−* is defined as the *sxtA4* gene not detected via single-cell multiplex PCRs and confirmed via additional screenings. These data demonstrated that the *sxtA4+* genotype dominated at all sites but showed a substantial difference between Florida and Puerto Rico, with a lower *sxtA4+* frequency from MB. It is worth noting that these were the first data demonstrating the presence of *sxtA4* in *P. bahamense* populations from Puerto Rico. These blooms are typically not tested for toxicity, and the general assumption is that they are non-toxic. These data suggested some cells within the population lack the genetic capacity for STX biosynthesis. However, these data were collected from various sites on individual days, not throughout the course of a season, which would be beneficial in examining toxin and overall bloom dynamics in *P. bahamense* populations. 

Many theories have been proposed as to the function of the toxin in the ecology of the species that produce it, with the leading hypothesis being that it functions as a grazing deterrent [21]. Numerous studies have been conducted with organisms from multiple trophic levels on the effects of STX as a grazing deterrent, with a range of results pertaining to the effects on health and active rejection of the toxic cells [22,23,24,25,26,27,28]. Being unicellular, the net growth rate in dinoflagellates is a balance of the cell division rate and the mortality rate due to grazers [29]. Copepods and their chemical cues are well-documented to induce toxin production [30,31]. However, chemical defenses are costly. Costs typically take the form of reduced physiological efficiency, manifested as mortality or slowed growth [32]. Leading up to bloom initiation, competition with co-occurring phytoplankton species along with high grazing pressure may select for toxin production. However, as the bloom develops, how and if toxicity changes remain unknown. For *P. bahamense*, its bioluminescence should also be considered, as bioluminescence in dinoflagellates also functions as a defense from predators, though flash function appears to differ between toxic and non-toxic species [1]. 

This study sought to gain insights into toxin dynamics among *P. bahamense* populations in the Indian River Lagoon, FL. The goals were to determine if and at what level *P. bahamense* populations produced toxins; to determine if differences occurred among spatially and temporally separate populations in Florida; and to obtain insights into the potential molecular mechanisms underlying this process. Here, we performed *sxtA4* genotype analysis from multiple sites in the IRL during a season. We also sampled consecutive (3–4) days over several months in 2023 at one specific location. We collected data on cell abundance, toxin quota per cell, *sxtA4* genotyping (2022, among multiple sites), and *sxtA4* and RubisCo (*rbcL*) transcription, for which we developed a *P. bahamense*-specific *rbcL* quantitative PCR assay for *rbcL* transcript measurements. Based on the collective results of toxin quota, genotyping, and transcriptional activity, a theory is presented to explain how the combined traits of toxicity and bioluminescence may be used for *P. bahamense* bloom development. 

## 2. Results

### 2.1. Field Sampling Sites and Parameters Measured

The Indian River Lagoon (IRL) is a shallow-water estuary that spans 156 miles along the east coast of Florida, from Ponce de Leon Inlet in Volusia County to Jupiter Inlet in Palm Beach County, covering ca. 40% of the coast. *P. bahamense* forms recurring annual blooms in the northern IRL. Water samples were collected from four sites (Figure 1) over the course of one year (2022) in the northern IRL. The four sites were distributed among the three water bodies that comprise the IRL and were termed as follows: 1. Diamond Bay (“DB”, located in the Banana River); 2. Haulover Canal (“HC”, Indian River); 3. Kelly Park (“KP”, Banana River); and 4. Beacon 42 (“B42”, Mosquito Lagoon). In general, the four sites were sampled on 2–3 consecutive days, though not the same dates for each site. In 2023, efforts focused on DB, with sampling 3–4 consecutive days in March, April, June, and July. For both seasons, the following parameters were measured (described in detail below): cell abundance; toxin quota per cell; *sxtA4* transcripts and *rbcL* transcripts (both normalized to per cell); the *sxtA4/rbcL* ratio; and for 2022, among the four sites, *sxtA4* genotype frequencies. 

### 2.2. P. bahamense Presence and Abundance 

Sampling among the four sites showed that presence and abundance vary spatially and temporally (Figure 2A,B). A detailed description of cell abundance is provided in the Appendix A.

*P. bahamense* is bioluminescent at all four sites. As with other dinoflagellate blooms, heterogeneity was observed among dates and sites (Figure 2B), though sampling occurred at the same location for each site, and typically within a 3 h mid-day window (11:00–14:00).

When sampling exclusively at DB over consecutive days over four months in 2023, as anticipated, there was a significant difference between March and all other months’ cell abundances (Figure 3A); also as anticipated, cell abundance fluctuated on a daily basis (Figure 3B). Sampling over multiple, consecutive days highlighted the oscillations in *P. bahamense* presence, be it from bloom growth/decline, cell transport via physical mechanisms, or differences among a 3 h sampling window.

### 2.3. Toxicity

When taking the collective average from multiple samplings spanning several months, KP displayed the highest average, at ~11.5 pg cell^−1^, though it was in a similar range to DB (~10 pg cell^−1^) and HC (~10.5 pg cell^−1^) (Figure 4A). This was in contrast to the average toxin quota per cell at B42, which was much lower, at 5.6 pg cell^−1^, than the other sites. Sampling the different sites over the course of a season illustrated fluctuations in toxicity (Figure 4B). 

DB was sampled exclusively over multiple consecutive days in 2023 (Figure 5A). Assimilating the average and individual toxin quota per cell for each month highlights the dynamic nature of cellular toxicity, as illustrated by the range of individual values, particularly in March and April (Figure 5B).

### 2.4. Relationship between Cell Abundance and Toxicity

We examined the relationship between cell abundance and toxicity, as we observed a trend in the toxin quota decreasing as cell abundance increased (Figure 6 and Appendix A). Spearman Rank Correlation analysis was applied to the combined total of all sites and dates from 2021 to 2022 (*n* = 27). A statistically significant negative correlation between cell abundance and toxin quota per cell was confirmed (Spearman Correlation coefficient, −0.434, *p* = 0.02). For DB 2023, the average cells per L and toxin quota per cell from 3–4 consecutive days of each month (March, April, June, July) was used. When including all four months, a moderate negative correlation was found to exist between cell abundance and toxin quota (Spearman Correlation coefficient = −0.4), though it was not statistically significant, likely because of the small sampling size (*n* = 4). When excluding the March sampling, in which samples did not pass the test for normality, a very strong (−1) negative correlation was found to exist between cell abundance and toxin quota.

### 2.5. sxtA4+ Genotyping

Single-cell *sxtA4* genotype analysis was performed on subsets of cells from the four IRL sites in 2022. Overall, *sxtA4+* was the dominant genotype at all sites and seasons though its frequency fluctuated temporally among sites (Figure 7).

Considering genotype frequencies in relation to cell abundance, the following trend was noted at the HC and KP sites: the *sxtA4+* genotype frequency was greater with lower cell abundance, while the *sxtA4−* genotype emerged with increased cell abundance (Figure 8B,C). DB displayed substantial variability. The *sxtA4−* genotype showed a trend of emerging as the season progressed, in which it was recorded at a higher-than-usual frequency in June (~35% on 23 June 22) and Sept 2022 (Figure 8A), during prime bloom season. For B42, unlike the other sites, the *sxtA4−* genotype was consistently detected (Figure 8D). However, genotyping was only performed on samples collected during late spring and summer. It is worth noting that when normalized to the number of samplings per site, the *sxtA4−* genotype occurred more frequently at B42 and HC, two locations that are particularly known for their strong bioluminescence. 

### 2.6. SxtA4 as Molecular Proxy for Toxicity

Collectively, *sxtA4* transcript levels mirrored those of toxicity, both averaged monthly and on an individual basis (Figure 9). Spearman Rank Correlation Analysis of average (monthly) *sxtA4* transcripts and toxin quota per cell from DB 2023 samples demonstrated a strong positive correlation (n = 4, r = 0.8). This supports the notion of *sxtA4* being used as a molecular proxy for toxin synthesis. On an individual date basis, *sxtA4* transcripts showed a moderately strong yet statistically insignificant positive correlation with toxin quota per cell (n = 13, r = 0.5, *p* = 0.08). Spearman Rank Order Correlation was also applied to 2022 samples, which had both *sxtA4* transcript measurements and toxin quota per cell; this resulted in a cumulative n of 11: 5 at DB, 1 at HC, 3 at KP, and 2 at B42. As with the 2023 samples, a positive correlation was found to exist between *sxtA4* transcripts and toxin quota (0.4), though because of the high variability, it was not statistically significant. However, it further confirmed the trend in *sxtA4* as a molecular proxy for toxin presence. 

An exact linear relationship between these two parameters should not be expected, as dinoflagellate genes are regulated more at the post-translational rather than transcriptional level [33,34,35,36,37]. Instead, these values should be taken as a qualitative rather than quantitative measure of toxicity. However, the fluctuations in transcripts on a daily basis in relation to toxin quota per cell suggest that not all cells express *stxA4*. Nevertheless, since these data were normalized to a per-cell basis based on cell abundance measurements, and some cells were shown to be *sxtA4−*, whether this is a reflection of genotype frequencies or overall toxin biosynthesis regulation cannot be answered by these data. 

Overall, whenever samples, including those of 2022, were collected for transcript measurements, *sxtA4* was detected (Appendix A), indicating that at least some cells in that population produced toxins. Additionally, Spearman Rank Correlation Analysis of *sxtA4* transcripts and cells per L from DB in 2023, using the average *sxtA4* transcript and cells per L values for each month (*n* = 4), showed a very strong negative correlation (−0.8) (although not statistically significant). This trend is anticipated because *sxtA4* served as a molecular proxy for toxin production. As toxin quota decreased with increased cell abundance, one would expect the transcripts to display a similar trend. 

### 2.7. rbcL qRT-PCR Assay

A *P. bahamense*-specific assay based on the RuBisCO large subunit (*rbcL*)was developed and optimized for use on field samples. The *rbcL* assay yielded a single peak via melt curve analysis on all eight *P. bahamense* strains tested. It did not amplify DNA from the dinoflagellates *Amphidinium carterae*, *Karlodinium veneficum*, *Karenia brevis*, *Alexandrium monilatum*, or *Pyrocystis fusiformis*; the diatom *Skeletonema costatum*; the green alga *Chlamydomas reinhardtii*; or any of the marine bacteria including *Roseobacter* spp., *Alteromonas macleodii*, *Vibrio alginolyticus*, or *Silicibacter* TM1040. The assay characteristics were slope = −3.4, y-intercept = 39.189, r^2^ = 0.978, and PCR efficiency = 96.42%, and it was able to detect down to 50 copies reliably (Appendix A). The assay was then applied to the same field samples as for *sxtA4* for use as a gauge of overall population growth/transcriptional activity and also as an assessment of overall RNA extraction success—i.e., samples with a very high C_T_ for *rbcL* yet which had adequate cell abundance, were discarded from further transcriptional analysis with *sxtA4*. This was the case with two samples from HC in 2022. 

### 2.8. rbcL qRT-PCR Assay to Assess Overall Population Activity

A very strong statistically significant positive correlation was found to exist between *sxtA4* transcripts and *rbcL* transcripts for both 2022 (r = 0.989, *p* = 0.0000002, n = 15) and 2023 (0.967, *p* = 0.0000002, n = 13). *RbcL* can serve as a molecular proxy for carbon fixation activity in marine phytoplankton [38,39], with a strong relationship shown to exist between *rbcL* transcriptional levels and photosynthetic activity. Though dinoflagellate genes in general are regulated more at the level of translation, and the Rubisco enzyme itself is subject to several mechanisms of metabolic regulation [40]. It was employed here to qualitatively assess the photosynthetic activity of the population and serve as a means to gauge *sxtA4* dynamics in relation to growth. In general, *sxtA4* transcription consistently displayed a similar pattern to *rbcL* in both 2022 (Appendix A) and 2023 (Appendix A), though overall transcript abundance was much lower.

### 2.9. Relationship between sxtA4 Transcripts and the sxtA4/rbcL Transcript Ratio

The relationship between *sxtA4* transcripts and the *sxtA4/rbcL* transcript ratio for the 2023 data was also examined. A strong, statistically significant negative correlation (−0.692, *p* = 0.008) was found between *stxA4* transcripts and the *sxtA4/rbcL* ratio for each individual sampling in DB 2023 (n = 13) (Figure 10). Comparing the sxtA4 transcripts per cell with the sxtA4/rbcL ratio and toxin quota per cell can provide insights as to potential underlying mechanisms of bloom toxicity: a low ratio yet with higher *sxtA4* transcripts and higher toxin per cell suggest that fewer cells within the population produce toxins (for example, data point 18 March 23). This was confirmed by a negative correlation between *sxtA4* transcripts and the *sxtA4/rbcL* ratio. We hypothesize this negative correlation means that though *sxtA4* and *rbcL* transcripts have a positive correlation and show similar patterns of regulation (Appendix A), *rbcL* transcripts increase at a faster rate than *sxtA4* transcripts. This suggests that (1) cells make fewer *sxtA4* transcripts or (2) more *sxtA4−* cells are present (which would result in fewer *sxtA4* transcripts since the gene is lacking in a portion of the population). However, the exact mechanism of toxin decrease—i.e., whether it is an increase in *sxtA4−* cells; *sxtA4* repression by *sxtA4+* cells; or a combination of both—within the overall population cannot be answered by these data. 

## 3. Discussion

Dinoflagellate species that form some of the most extensive toxic harmful algal blooms (HABs) are also bioluminescent, yet these two traits are rarely linked when considering the ecological significance of either [1]. Both traits are energetically expensive, and both are proposed to function as a defense from predators. Why these *P. bahamense* populations harbor both, and their roles in bloom persistence, remains unanswered. The trend in toxin quota per cell decreasing as cell abundance increased is an intriguing find, and here the role of bioluminescence in *P. bahamense* bloom development should be considered with toxicity. Bioluminescence and toxicity differ among species. A general trend seems to be that the smaller “dim emitters”, such as *Lingulodinium polyedra* (yessotoxin) and *P. bahamense*, are also toxic, while the larger “bright emitters”, such as *Pyrocystis* spp., are not [1]. For many years, the leading hypothesis was that the bioluminescent flash acted as a “burglar alarm” to attract predators (fish, cephalopods) of the dinoflagellate’s predator [41,42,43,44]. Since many bloom-forming species are also toxic, an alternative hypothesis is that the bioluminescent flash in toxic species functions as an aposematic warning, signaling unpalatability to grazers [1]. Previous work with toxic dim-emitter species showed it functioned as an aposematic signal at low cell concentrations and a burglar alarm at high cell concentrations [45,46]. An extension of aposematism is Batesian mimicry. Batesian mimicry evolves when individuals of a palatable, non-toxic species (the mimic) achieve the selective advantage of reduced predation because of their resemblance to a toxic species (the model) that predators avoid [47,48]. Strong support for Batesian mimicry in dinoflagellates has come from experiments in the Widder lab that looked at copepod grazing on toxic and non-toxic bioluminescent dinoflagellates [45]. From these studies, the notion of Batesian mimicry in dinoflagellates arose, in which non-toxic dinoflagellates mimic the flash of unpalatable (toxic) species, thereby benefitting from the learned avoidance behavior of the predator [49]. Automimicry is Batesian mimicry within the same species, in which the mimics benefit from the learned avoidance of the predator but pay no individual cost of toxicity themselves [32,50]. When considering our results with *P. bahamense* field populations, the decrease in toxin production coupled with the two different genotypes indicates an automimicry-like process may occur during bloom development. 

We propose a hypothesis based on cell abundance, such that all or nearly all cells are toxic and bioluminescent at the start of the bloom. At low cell concentrations, the flash serves as an aposematic signal, warning predators of unpalatability. A portion of the population then loses toxicity but keeps bioluminescence (=automimics?), allowing this portion of the population to grow more quickly and thus expand the bloom. At high cell concentrations, bioluminescence functions as a burglar alarm. Toxicity is not needed for the burglar alarm. Our lab studies provide solid evidence that toxicity and bioluminescence are linked, and this link is based on cell abundance as related to flash function. The field data for *P. bahamense* suggest a similar process. However, these are small numbers and did not capture full-bloom events. Additionally, the inverse correlation between cell density and cell toxin quota may reflect a concentrating or diluting factor when populations grow slowly or quickly, respectively, and so must also be considered as an alternative to explain the trend in *P. bahamense* bloom toxicity.

The molecular mechanisms underlying the decrease in toxicity in these blooms remain to be determined, specifically, whether it occurs at the genetic or transcriptional level. It may be an increase in *sxtA4−* genotype frequency; gene repression in *sxtA4+* cells; or a combination of both. Overall, *sxtA4* was the dominant genotype at all locations, as would be expected in automimicry, though the percent frequency varied spatially and temporally. Ideally, this analysis would be performed early in the season among sites for multiple days. The role of transcription must be considered. While *sxtA4* has been found in all toxic strains and species, the gene has also been found in several non-toxic and/or mutant strains of toxic species. However, no expression of *sxtA4* at the mRNA level has been detected in the non-toxic strains [10,12,16,51]. *SxtA4* transcript levels were measured as a means to assess population transcriptional activity using a modified qPCR that accounted for sequence differences in *P. bahamense* in the region targeted by the reverse primer. Since *sxtA4* has been shown to be necessary for toxin biosynthesis, it can serve as a molecular proxy for toxin production. Here, *sxtA4* transcripts had a positive correlation with toxin quota. It was not a linear correlation, nor was it expected to be, as numerous studies indicate dinoflagellate gene expression is controlled more at the level of translation rather than transcription [33,34,35,36,37]. Rather, it provided a qualitative means of assessing toxin biosynthesis activity at the population level. *SxtA4* transcripts were found to have a negative correlation with cell density. The *P. bahamense*-specific *rbcL* assay was then developed for use on the same field samples. *RbcL* can serve as a molecular proxy for carbon fixation activity in marine phytoplankton [38,39], with a strong relationship shown to exist between *rbcL* transcriptional levels and photosynthetic activity. Though dinoflagellate genes in general are regulated more at the level of translation, and the Rubisco enzyme itself is subject to several mechanisms of metabolic regulation [40], it was employed here to qualitatively assess the overall transcriptional activity of the population, as *P. bahamense* is photosynthetic and samples were collected during the day. It also served as a means to gauge *sxtA4* transcriptional dynamics. *SxtA4* normalized to *rbcL* is an indicator of the toxin biosynthesis activity of the population at that time point. However, a negative correlation was found between *sxtA4* transcripts and the *sxtA4/rbcL* ratio, meaning that as *sxtA4* transcripts increased, their abundance relative to *rbcL*, and thus overall activity, decreased. As *rbcL* is an indicator of activity, this suggests as the bloom expands, toxin biosynthesis decreases. This model based on qualitative transcriptional activity is in agreement with the negative correlation observed for the phenotypic measurements of cell density and toxicity, with toxicity decreasing with increased cell abundance. 

Of note is that these measurements are all derived from the PSP ELISA kit. There were several samplings at KP in March 2023 in which *sxtA4* transcripts were detected but not toxins. While quantitative, this ELISA is primarily specific for STX (and cross-reacts with two of the other congeners in small percentages). It does not account for all the other congeners. The fact that *sxtA4* was detected but not STX suggests the early-season KP populations have a toxin profile different from that of DB, of which STX is not a primary component. The nitrogen source has been found to affect toxin profiles in *P. bahamense* isolates from the IRL [52]. Different N species and their concentrations were not measured here, but this underscores the need to measure not just total toxin but also the congener profiles among *P. bahamense* populations when considering the factors underlying bloom success and expansion. 

Our future plans include bioluminescence measurements, normalized to a per-cell basis, at sites where toxicity is measured. Our future research also includes quantifying *sxtA4* gene copy number commitment with *sxtA4* transcript copy number as a means to gain further insights as to whether changes in bloom toxicity are driven more at the genetic (emergence of *sxtA4−* cells) or transcriptional (repression of *sxtA4* in *sxtA4+* cells) level. Both traits are energetically expensive [1]. In Puerto Rico, home of three bioluminescent bays in which *P. bahamense* serves as the source of bioluminescence, toxicity has not been reported (though the *sxtA4* gene has been found [20]). The reason for Florida populations harboring (and expressing) two energetically expensive phenotypes suggests a linkage, with both traits potentially serving a role in bloom formation, though this link remains to be identified. The results of the single-cell genotyping showed that the *sxtA4−* genotype occurred more frequently (or always) at B42 and HC, two locations that are particularly known for their strong bioluminescence. How and if this is significant in the ecology of *P. bahamense* is not known, but these data further suggest a link between these two traits. 

Following the saxitoxin pufferfish poisoning outbreaks across the United States in 2002–2004, and the realization that *P. bahamense* was the source, the Florida Wildlife Commission (FWC) imposed a permanent ban on puffer fish harvesting in the IRL. At this same time, STX was detected in shellfish from the IRL, and the Florida Department of Agriculture and Consumer Services (FDACS) and the FWC established the PSP Biotoxin Contingency Program in April 2003 to monitor water for *P. bahamense* and shellfish for saxitoxin. Additionally, the FWC implemented the Red Tide Offshore Monitoring Program, for not only *P. bahamense* but other HAB species (URL https://myfwc.com/research/redtide/monitoring/ accessed on 16 June 2024). Right now, a lot of that work is handled by volunteers. Our efforts can help secure more volunteers and increase sampling coverage. Additionally, the molecular tools described in this study can be used in conjunction with sampling efforts to determine the extent of, toxicity in these *P. bahamense* populations. 

## 4. Materials and Methods

### 4.1. Sample Collection 

Whole water samples were collected from the surface to a depth of 0.25 m and concentrated by size-fractionating 10–25 L water through 30-micron mesh. The biomass was rinsed into a sterile 500 mL Nalgene bottle using filtered water from that site. Multiple replicates were collected for dedicated RNA extraction; toxin measurements and cell counts; and single-cell isolation. *SxtA4* and *rbcL* transcript abundance was measured beginning in June 2022. Typically, 10–15 L was collected per sample for RNA extraction. Samples collected for single-cell isolation and genotyping were typically collected as 5 L samples, and filtered water was added to the bottle for transport. Samples were transported at ambient temperature back to the lab because transport on ice resulted in stressed cells, as indicated via cell morphology, motility, and pigmentation (Cusick, pers. obs). Concentrated water samples were filtered under low vacuum. Samples for RNA extraction were collected onto Nucleopore (1 µm) polycarbonate filters. Depending on biomass, 1–3 filters were needed per RNA sample. All filters from the same RNA sample were combined into a single microcentrifuge tube, covered in RNALater, and incubated for ~5 min at room temperature before storage at −20 °C. Samples for toxin analysis were filtered onto Whatman AE filters and stored at −20 °C until extraction. Approximately 5 mL of a filter-concentrated sample was preserved with 2% glutaraldehyde and stored at 4 °C for cell enumeration. 

### 4.2. Cell Enumeration

*P. bahamense* abundance was determined via cell counts from preserved field samples using a Sedgewick Rafter counting chamber with standard light microscopy under 200×, 400×, and 1000× magnification based on morphological features previously defined in the literature [53]. Triplicate cell counts were performed for each sample. 

### 4.3. Toxin Analysis

STX was extracted with milli-Q water containing 1% acetic acid (*v*/*v*). The filter was combined with 5 mL extraction solvent. The combined solution was freeze-thawed three times to extract STX. Extracts were clarified by centrifugation at 16,000× *g* for 10 min and passed through a 0.45 µm nylon syringe filter. The supernatant was analyzed in duplicate using a commercial Saxitoxin (PSP) ELISA kit (Eurofins Abraxis LLC, Warminster, PA, USA) to determine the STX concentration according to the manufacturer’s instructions. Appropriate dilutions were applied to each sample to fit the standard curve (0–0.4 ppb). 

### 4.4. sxtA4 Genotype Analysis

#### 4.4.1. Cell Isolation and Lysis

Methods for single-cell isolation, cell lysis, and subsequent PCRs used methods developed previously for single-cell *P. bahamense* genotype analysis [20,54], with slight modifications from some collections to include more vigorous washing of cells, as samples from some sites presented challenges likely because of contaminants. Briefly, an aliquot of a water sample was reverse-filtered one–three times through clean 20 µm mesh or cell washers with 0.2 µm filtered water from the Indian River Lagoon. One milliliter aliquots were placed in a petri dish and viewed under light microscopy. *P. bahamense* cells were identified under 200×, 400×, and 1000× magnification based on morphological features previously defined in the literature [53]. Individual cells were then isolated via glass micropipette, washed in sterile nuclease-free water, placed in sterile 200 µL PCR tubes, and stored at −20 °C until cell lysis and PCR. Prior to cell lysis, each sample was brought to a volume of 20 µL by the addition of nuclease-free water. Cells were lysed via five consecutive freeze–thaw cycles of 30 s each alternating between a dry ice/ethanol slurry and heating to 98 °C in a thermocycler. Samples were then centrifuged and placed on ice, and PCR reagents were added directly to the tubes. 

#### 4.4.2. Multiplex PCR

The *Pyrodinium*-specific rRNA small subunit primers (Pcomp370F/Pcomp1530R) used previously did not amplify as consistently as in previous years. Therefore, the primer set 18ScomF1/Dino18SR1 (Table 1) [55], which amplifies an ~650 bp region of the 18S rRNA gene, was tested and optimized. Single-cell conditions were evaluated and optimized using a range of conditions, including magnesium, primer, and Taq concentrations; annealing temperature; and two different sets of PCR reagents including the GoTaq Flexi DNA polymerase kit (Promega. Madison, WI, USA) and the Titanium taq kit (TaKeRa Bio USA, San Jose, CA, USA). For the majority of samples, PCRs were conducted as follows (as final concentrations): 1× GoTaq Flexi buffer, 2 mM MgCl_2_, 0.3 mM dNTP mix, 400 nM each forward and reverse primer of both primer sets, 3.7 U GoTaq Flexi DNA polymerase, and brought to a final volume of 50 µL with nuclease-free water. Multiplex PCR was performed on single cells using the primer sets 18ScomF1/Dino18SR1 and *sxtA4*-specific primers sxtA4007/PyroRq (product size ~365 bp). Thermocycling conditions consisted of an initial denaturation at 95 °C for 3 min followed by 40 cycles of 95 °C for 30 s; 57 °C for 15 s; and 72 °C for 45 s with a final extension at 72 °C for 7 min. Products were visualized on 1% E-Gel EX agarose gels with the E-Gel Power Snap Electrophoresis System (Invitrogen, Carlsbad, CA, USA). The 18S rRNA gene served as a positive control to confirm the presence of the cell in the tube and that reactions were not inhibited by potential contaminants. Only samples yielding the 18S rRNA gene amplicon were included in *sxtA4* genotype frequency analysis. 

Samples yielding only a band indicative of the 18S rRNA gene were PCR-purified and sequenced with 18ScomF1 to confirm *P. bahamense* specificity. Subsequent PCRs were then performed on all samples, which yielded only the 18S rRNA gene amplicon to further confirm the presence/absence of *sxtA4* using stxA4166F/678R (product size ~500 bp, Table 1), in which all contents of the multiplex PCR tube (which included the lysed single cell) were cleaned using the MP GeneClean kit (MP Biomedicals, Santa Ana, CA, USA) following the protocol for genomic DNA. The resulting eluant, undiluted and at a 1:10 dilution, served as the template (2 µL). PCRs utilized the GoTaq PCR Core System I and consisted of the following (as final concentrations): 1× Colorless GoTaq buffer (which included as final concentration 1.5 mM MgCl_2_), 0.1 mM dNTP mix, 400 nM each sxtA166F and sxtA678R, and 0.625 U GoTaq DNA polymerase brought to a final volume of 25 µL with nuclease-free water. Thermocycling conditions consisted of 95 °C for 3 min, followed by 35 cycles of 95 °C, 30 s, 57 °C, 15 s, 72 °C, 40 s, and a final extension at 72 °C for 7 min. The resulting product was visualized with gel electrophoresis. Samples for sequencing were first cleaned using the QIAquick PCR Purification Kit (Qiagen, Germantown, MD, USA) and prepared following standard protocols of Azenta for pre-mixed samples. Samples were sequenced (Azenta, Germantown, MD, USA) using the 18ScomF1 (for the partial 18S rRNA gene), 007, or sxtA4166F (for *sxtA4*) primer. To confirm their identity, the obtained sequences of *sxtA4* and the 18S rRNA gene were queried against the GenBank nr database using standard BLASTN 2.2.26+ [57].

### 4.5. RNA Extraction 

Total RNA was extracted using the RNeasy Biofilm or RNeasy Microbiome kit (Qiagen) with slight modifications to the manufacturer’s protocol. Filters were removed from the centrifuge tube using sterile forceps. Excess RNALater was removed from the filters by pressing the forceps against the filters while in the centrifuge tube. Filters were placed in the bead tube. The microcentrifuge tube minus the filters was centrifuged at 10,000× *g* for 5 min, and any residual RNALater was removed with a pipet tip. Any biomass remaining was re-suspended in the lysis buffer, and the entire aliquot was transferred to the bead tube containing the filters. The samples were subject to three rounds of bead beating at 8 m/s for 40 s using an MP Biomedical Fastprep bead beater. The samples were chilled on ice for 45 s in between each round. The remaining steps followed the manufacturer’s protocol. Optional on-column DNase digest was performed on all samples, with the digestion time extended to 20 min. Samples were eluted in 50 µL RNase-free water and stored at −80 °C until cDNA synthesis. Total RNA concentration and purity were assessed using the Nanodrop 1C.

### 4.6. cDNA Synthesis 

Total RNA was converted to cDNA using the High-Capacity RNA-to-cDNA kit (Applied Biosystems, Foster City, CA, USA). Each reaction contained 10 µL 2× RT buffer, 1 µL 20× enzyme, and total RNA, brought to a final volume of 20 µL with nuclease-free water. The presence of inhibitors was assessed by setting up reactions from the same sites with 3 µL and 9 µL total RNA, and the C_T_ values were compared. In most cases, 3 µL yielded lower C_T_s. No-RT controls consisted of all components except the enzyme. The reactions were performed in the ProFlex thermocycler (Applied Biosystems, Carlsbad, CA, USA) under the following conditions: 37 °C for 60 min, followed by 95 °C for 5 min. 

### 4.7. sxtA4 Quantitative Reverse Transcription PCR Assay 

A qRT-PCR assay previously developed for the detection and quantification of *sxtA4* transcripts in field samples containing mixed phytoplankton assemblages [56] was slightly modified as the comparison of primer sequences with *P. bahamense* sequences obtained from lab isolates and environmental clones showed a one bp difference in the reverse primer sequence, or, alternatively, the deletion of a codon in the *P. bahamense* sequence. The assay used the original forward primer (072F, CTTGCCCGCCATATGTGCTT) and the re-designed reverse (PyroRq, GCCGCCGCCACCATATCC) primer (Table 1). Product specificity from both field and lab samples was confirmed via melt curve analysis and sequencing. Reactions were performed with the PowerUp SYBR Green Master Mix (Applied Biosystems, Foster City, CA). Each reaction contained 10 μL 2× PowerUp SYBR Green master mix, 300 nM of each forward and reverse primer, and 2 μL of template cDNA, brought to a final volume of 20 μL with nuclease-free water. No-RT controls consisted of all components except the enzyme. The reactions were performed on the QuantStudio 6 Real-Time PCR System with the 96-well block format (Applied Biosystems, Foster City, CA, USA). The following protocol was used for all reactions: an initial 20 s incubation at 95 °C, followed by 40 cycles of 95 °C for 1 s and 60 °C for 20 s, followed by a melt curve analysis of 95 °C for 15 s, 60 °C for 1 min, and 95 °C for 15 s to determine product specificity. All qPCR reactions were performed in duplicate using Applied Biosystems MicroAmp Fast 96-well reaction plates sealed with MicroAmp optical adhesive film. No-template controls were also included in each amplification run to monitor for contamination. Reactions were recorded and analyzed using Applied Biosystems QuantStudio 6 System software. An eight-point standard curve was constructed from a purified PCR product of the nearly full-length *sxtA4* gene amplified from a lab isolate and used for absolute quantification. The number of copies per µL was calculated using the equation X g µL^−1^ (DNA/[PCR amplicon × 660]) × 6.022 × 1023. The PCR product stock solution was diluted in nuclease-free water to a working concentration of 5 × 10^8^ copies per µL. Serial 10-fold dilutions were used to construct a standard curve spanning eight orders of magnitude from 5 × 10^7^ to 5 × 10^−1^ copies per µL. In total, 2 µL of each dilution was used per reaction, yielding a standard curve that ranged from 1 × 10^1^ to 1 × 10^8^ copies. The assay characteristics were slope = −3.474, y-intercept = 40.782, r^2^ = 0.998, and PCR efficiency = 94.03%. Data were recorded as total transcripts in each 20 µL reaction; these values were then normalized to transcripts per cell based on cell counts of that sample with the volumes of RNA extract, cDNA synthesis, and volume in qPCR reaction accounted for. 

### 4.8. Development of rbcL Quantitative Reverse Transcription PCR Assay

A *Pyrodinium*-specific quantitative reverse transcription PCR assay targeting the RuBisCo large subunit was developed for use with SYBR Green chemistry. As samples were collected during daylight hours, and *P. bahamense* is photosynthetic, this assay was developed to serve as a proxy in gauging the overall transcriptional activity of the population, therefore serving as a qualitative assessment of *sxtA4* transcriptional activity (i.e., if *sxtA4* transcripts were low, it would aid in determining if it was because only a few cells expressed the gene or due to low overall activity of the population). 

#### 4.8.1. *P. bahamense rbcL* Sequence Analysis

The *P. bahamense rbcL* mRNA sequence (the only *rbcL* sequence data available for *P. bahamense* as of 03-10-2024) (TR41270_c2_g2_i1) was downloaded from the MMETSP dataset available through the CAMERA Data Distribution Center (URL http://camera.crbs.ucsd.edu/mmetsp/, accessed on 10 March 2024). Primers were manually designed to target the 5′ and 3′ ends of the transcript to obtain the longest sequence possible. Multiple primer sets were designed and tested on both the DNA and cDNA of a lab strain isolated from the IRL. PCRs utilized the GoTaq PCR Core System I and consisted of the following (as final concentrations): 1× Colorless GoTaq buffer (which included as final concentration 1.5 mM MgCl_2_), 0.1 mM dNTP mix, 400 nM each forward and reverse primer, 0.625 U GoTaq DNA polymerase, and 2 µL template, brought to a final volume of 25 µL with nuclease-free water. Thermocycling conditions consisted of 95 °C for 3 min, followed by 35 cycles of 95 °C, 30 s, 49–54 °C (depending on primer set), 15 s, 72 °C, 1 min, with a final extension at 72 °C for 7 min. Gel electrophoresis showed bands of identical size for both DNA and cDNA PCRs, and so the reactions yielding the longest amplicon (997 bp; rbcLF1/R1, Table 1) were cleaned using the Qiagen QIAquick PCR Purification Kit and sequenced using the rbcLF1 primer. A comparison of these DNA and cDNA sequences showed they were nearly identical. No introns were detected. The *rbcL* cDNA sequence described here has been deposited in GenBank under Accession number PQ044389.

To identify regions within the *rbcL* sequence specific to *P. bahamense*, full-length or nearly full-length *rbcL* sequences from other dinoflagellate species were also downloaded from GenBank (last access 4 March 2024), in particular, those whose habitat includes the Indian River Lagoon. Sequences were aligned in MEGA7 using ClustalW. Primers were manually designed and analyzed for parameters such as annealing temperature, secondary structures, and potential dimerization using the Oligonucleotide Properties Calculator (http://biotools.nubic.northwestern.edu/OligoCalc.html, (accessed on1 June 2024)) and IDT OligoAnalyzer Tool. This resulted in three primer sets; these were first tested and optimized using DNA and cDNA from the lab strain (FWC001). The primers targeted an approximately 100 bp region of the coding sequence. Initially, primer concentrations of 200, 400, and 600 nm (as final concentrations) were evaluated with temperatures of 58, 60, and 62 °C using the PowerUp Fast SYBR Green master mix. The optimized annealing/extension temperature was found to be 60 °C, with a final primer concentration of 600 nM. All primer sets gave a single peak; the primer set with the lowest C_T_ value was selected for additional specificity and sensitivity testing. 

#### 4.8.2. Assay Specificity and Sensitivity

The *rbcL* assay was applied to genomic DNA from an additional seven *P. bahamense* strains, isolated from the Indian River Lagoon (Pyb16, Pyb21A, Pyb23) or Puerto Rico (ARC 432, ARC 433, ARC 434, ARC 438, Algal Resources Collection, University of North Carolina) to test for assay specificity using the conditions described above. The assay was also tested with other species of dinoflagellates, diatoms, and algae. These included the dinoflagellates *A. carterae*, *K. veneficum*, *K. brevis*, *A. monilatum*, and *P. fusiformis* (the latter two both common in the Indian River Lagoon [Indian River Lagoon Species Inventory, irlspecies.org]; Cusick, pers. obs.); the diatom *S. costatum* (also commonly found in the IRL [58]; and the green alga *C. reinhardtii*. The assay was also applied to several species of marine bacteria including *Roseobacter* spp., *A. macleodii*, *V. alginolyticus*, and *Silicibacter* TM1040. 

An eight-point standard curve was constructed from the 997 bp purified PCR product of the *rbcL* gene amplified from *P. bahamense* FWC001 and used for absolute quantification using the methods described above for the *sxtA4* assay. Serial 10-fold dilutions were used to construct a standard curve spanning eight orders of magnitude from 5 × 10^7^ to 5 × 10^−1^ copies per µL; an additional 6-point standard curve with 1:2 dilutions was used to determine the limit of detection of this assay. 

### 4.9. Statistical Analysis 

Statistical analysis was performed in SigmaPlot v14. The parameters of cell abundance; toxin quota per cell; *sxtA4* transcripts; *rbcL* transcripts; and *sxtA4/rbcL* transcript ratio were assessed for normality (Shapiro–Wilk) and equal variance (Brown–Forsythe). A *t*-test (student’s, or Welch’s if data did not pass the test for equal variance) or Mann–Whitney Rank Sum (if data did not pass the test for normality) test was used to test for significant differences in toxin quota per cell and cell abundance among sites (2022 data) and among months (DB 2023). For DB 2023 cell abundance comparisons, we used a one-tailed *t*-test as we hypothesized that cell abundance would increase between Mar and April, Mar and June, and Mar and July. Spearman Rank Order Correlation analysis was applied (as most data were not normally distributed) to examine relationships among different parameters. In general, for 2021–2022, all sites and dates were either (1) analyzed as a collective group (for correlation analysis) or (2) among sites (*t*-tests). For DB 2023, samples were either analyzed as a collective group (typically, correlation analysis) or among months (*t*-tests).

## Figures and Tables

**Figure 1 marinedrugs-22-00311-f001:**
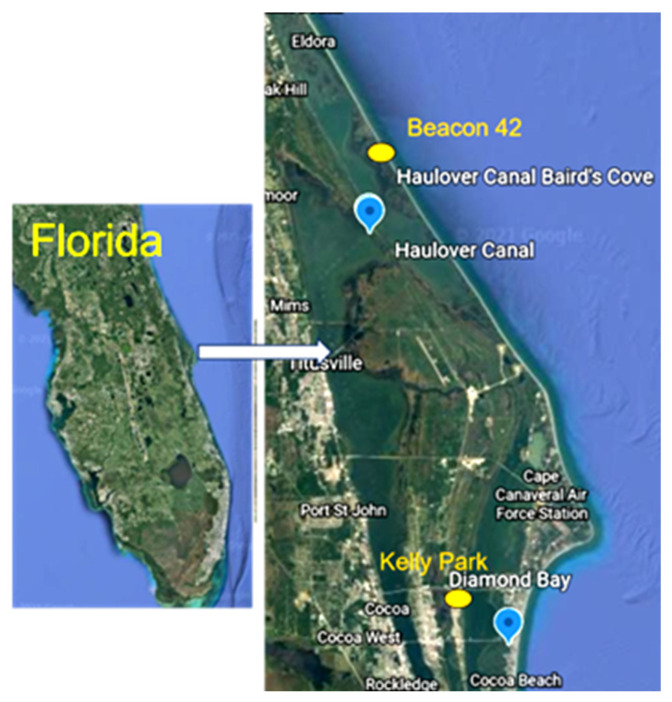
Four sampling sites in the Indian River Lagoon. Blue dots illustrate the locations of Haulover Canal (located in the Indian River) and Diamond Bay (located in the Banana River); sampling was initiated at these two sites in 2021. Yellow ovals indicate the locations of Beacon 42 (located in the Mosquito Lagoon) and Kelly Park (located in the Banana River), two sites added in 2022.

**Figure 2 marinedrugs-22-00311-f002:**
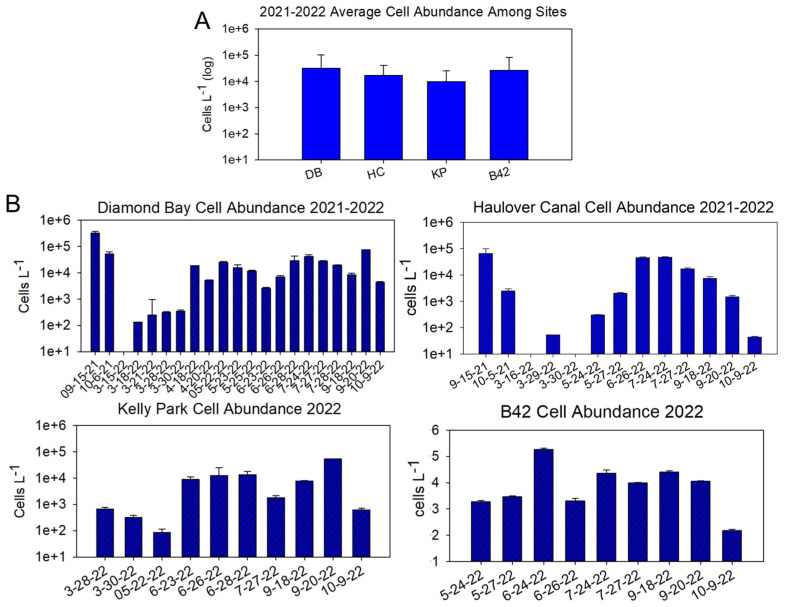
(**A**) Average *P. bahamense* cell abundance at four sites in the Indian River Lagoon over the course of a year. (**B**) Daily cell abundance at each site. *P. bahamense* abundance was determined via cell counts from preserved field samples using a Sedgewick Rafter counting chamber with standard light microscopy. Cell counts were performed in triplicate. Note that sampling dates are not the same across all sites.

**Figure 3 marinedrugs-22-00311-f003:**
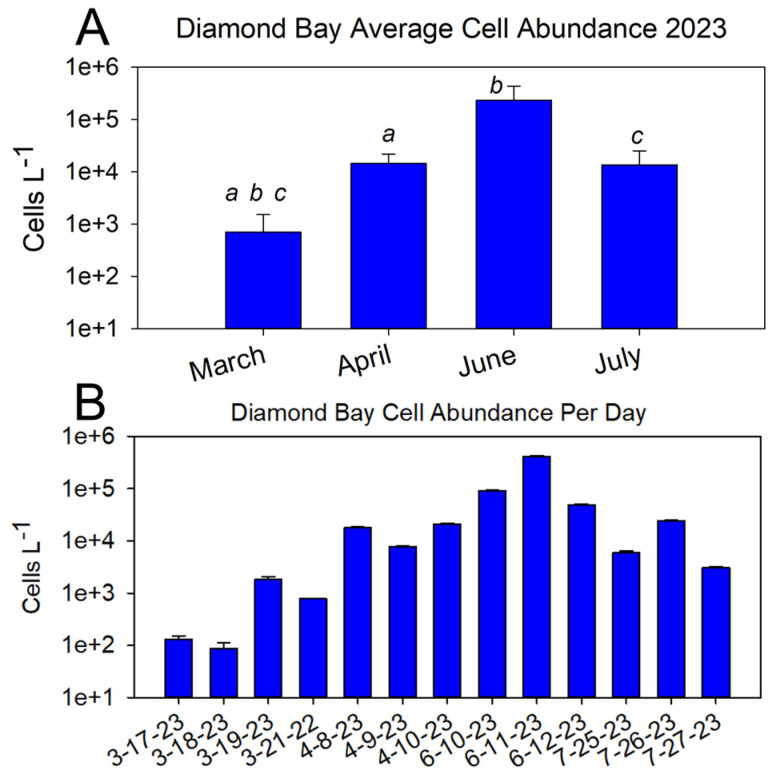
(**A**) Average *P. bahamense* cell abundance from site DB over multiple months spanning early and prime bloom seasons. Cell counts were performed in triplicate from each sampling using a Sedgewick Rafter counting chamber for 2022 samples. a, b, and c indicate significant differences among months (*p* < 0.05, one-tailed Student’s *t*-test): a = significant difference between March and April; b = significant difference between March and June; c = significant difference between March and July. There were no significant differences in cell abundance among the other months. (**B**) Daily cell abundance at each site.

**Figure 4 marinedrugs-22-00311-f004:**
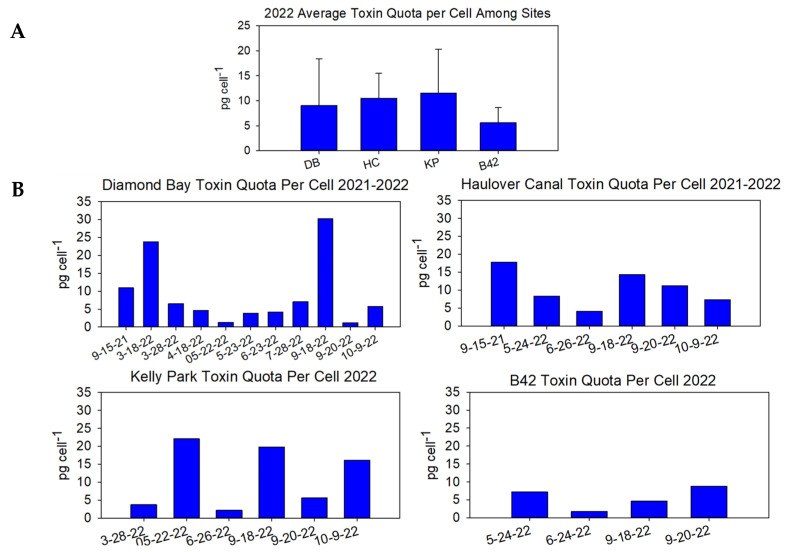
(**A**) Average toxin quota per cell among the four sites in 2022. (**B**) Daily toxin quota per cell among the different sites. Toxicity was determined via the Abraxis PSP ELISA and normalized to a per-cell basis based on cell abundance measurements. Note that Diamond Bay and Haulover Canal also included measurements from 2021.

**Figure 5 marinedrugs-22-00311-f005:**
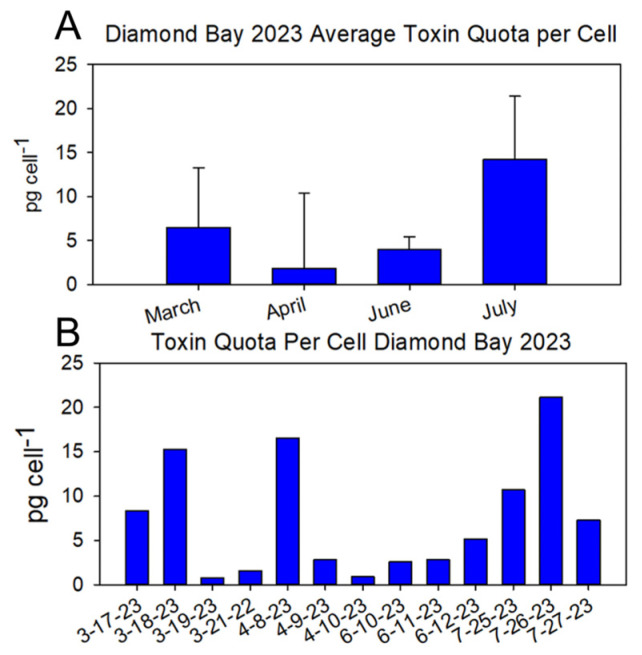
(**A**) Average and (**B**) daily toxin quota per cell at Diamond Bay in 2023. Toxicity was determined using the Abraxis ELISA PSP kit and normalized on a per-cell basis based on cell abundance measurements. The average of each month was calculated from the cumulative samplings for that month (*n* = 4, March; *n* = 3, April, June, July).

**Figure 6 marinedrugs-22-00311-f006:**
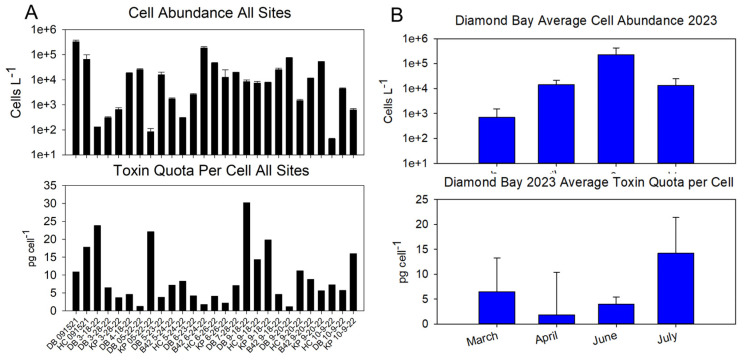
(**A**) Profile of cell abundance and corresponding toxin quota per cell among all four sites in 2022. (**B**) Profile of average cell abundance and corresponding toxin quota per cell at DB in 2023. (Profile of DB daily values as Appendix A).

**Figure 7 marinedrugs-22-00311-f007:**
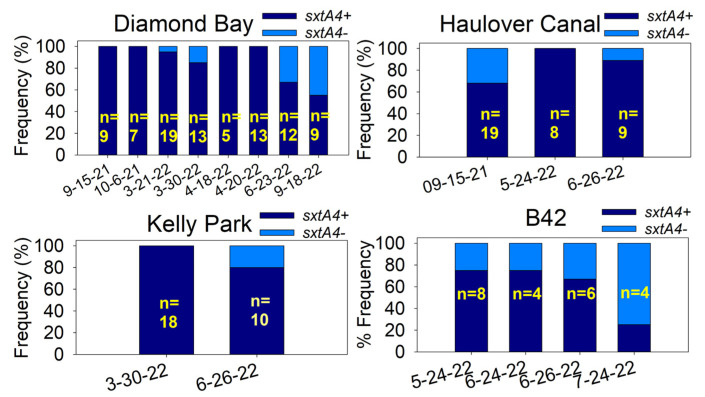
*SxtA4* genotype frequencies on different dates throughout the season among the four sites. n = number of cells analyzed.

**Figure 8 marinedrugs-22-00311-f008:**
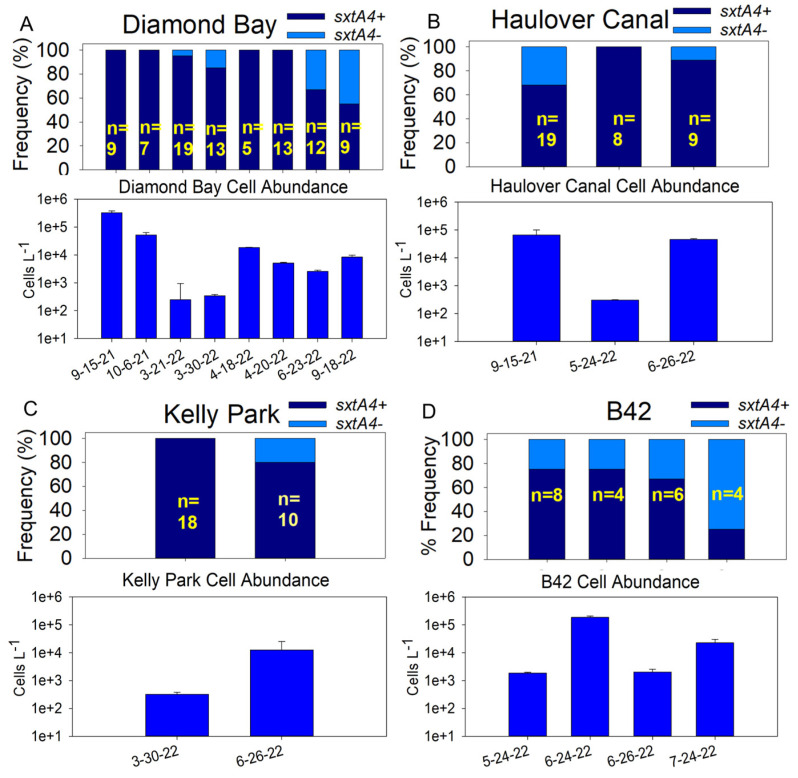
Genotype frequencies in relation to cell abundance among the four sites: (**A**) Diamond Bay; (**B**) Haulover Canal; (**C**) Kelly Park; (**D**) Beacon 42 (B42). n = number of cells analyzed via the single-cell multiplex PCR. The y-axis indicates % frequency with a total of 100%; *sxtA4−* is designated by a light blue portion of a bar; *sxtA4+* is designated by a dark blue portion of a bar.

**Figure 9 marinedrugs-22-00311-f009:**
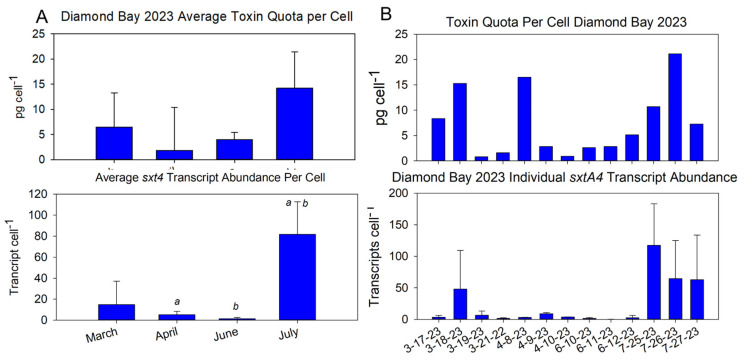
SxtA4 transcripts as a molecular proxy for toxin biosynthesis. *SxtA4* transcript abundance was quantified using the qPCR assay and normalized to a per-cell basis based on cell counts corresponding to that sample. (**A**) Diamond Bay average values for each of the four months and (**B**) Individual daily values over the four months.

**Figure 10 marinedrugs-22-00311-f010:**
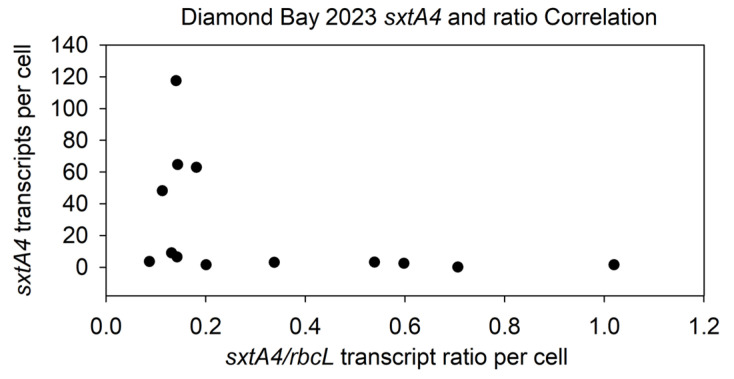
Correlation analysis of *sxtA4* transcripts and the *sxtA4/rbcL* transcript ratio from DB in 2023. A negative correlation was found between *sxtA4* transcripts and the *sxtA4/rbcL* ratio in DB 2023 samples (−0.692, *p* = 0.008) (*n* = 13).

**Table 1 marinedrugs-22-00311-t001:** List of primers used in this study.

Primer Name	Sequence	Source	Method/Notes
18ScomF1	GCTTGTCTCAAAGATTAAGCCATGC	[55]	Single-cell PCR; 18ScomF1/Dino18SR1
Dino18SR1	GAGCCAGATRCDCACCCA	[55]	
SxtA4166F	CAT GGC TGC GGC GTT CTT G	[20]	Follow-Up PCRs of single cells; 166F/678R
SxtA4678R	GAT GGG GTA CCA CAT AGG G	this study	
PyroRq	GCCGCCGCCACCATATCC	this study	PyroRq/sxt072 qPCR
sxt072	CTTGCCCGCCATATGTGCTT	[56]	
sxt007	ATGCTCAACATGGGAGTCATCC	[10]	Single-cell PCR; 007/PyroRQ
rbcLF1	ATGATGTGCTCGGTCCTAAC	this study	PCR of gDNA and cDNA; RbcLF1/R1
rbcLR1	CATCAGAGAGGCTCACATCA	this study	
rbcL186F	TGGCGAGGCCTGTTACGC	this study	rbcL186F/312R qPCR
rbcL312R	CGA GCC GGT CTC CTT GAT G	this study	

## Data Availability

The sequences obtained in this study are in the process of being deposited in GenBank and will be made freely available upon acceptance.

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
