# Peer review of "Toxin Dynamics among Populations of the Bioluminescent HAB Species Pyrodinium bahamense from the Indian River Lagoon, FL"

_marinedrugs, 2024, doi:10.3390/md22070311_

Round 1

Reviewer 1 Report

Comments and Suggestions for Authors

This manuscript by Cusick et al. details the correlation between bioluminescence, saxitoxin producing capabilities, and density of growth in Pyrodinium bahamense sampled from four regions along the Indian River Lagoon in Florida over the course of a year. One region was further sampled for an additional year. P. bahamense is not always toxic, and the dynamics of toxin production in their blooms were generally not known. The authors use several analytical and quantitative techniques to assess the presence of a portion of a saxitoxin biosynthetic gene, sxtA4, rubisco transcripts relative to growth, and saxitoxin presence. In general, they observe that the overall density of the bloom negatively correlates with saxitoxin and saxitoxin producing capabilities based on the sxtA4+/- genotype. The authors propose in the discussion that saxitoxin production as a feeding deterrent is more important in the early growth stages of the bloom, whereas bioluminescence is the dominant defensive property in more dense growth states. This is an interesting argument in terms of metabolic potential and bloom dynamics, particularly given that there are both sxtA4+ and sxtA4- genotypes present. I found the discussion section of this article to be quite nice and interesting to consider relative to the findings, which are scientifically sound. Overall, this is an interesting read with some minor edits described below:

There is only one portion where I think necessary data is absent. On page 8, in the rbcL qRT-PCR assay section, there are several parts that say "data not shown" but values are given for an equation that is fit to the data. I think the data should be shown in this case and it is unclear why it would not be shown if it is possible to fit an equation to it. In this section, I am also unsure how PCR efficiency is determined as this is not a parameter I am familiar with. 

In terms of the writing, there are some portions that I think could be made more clear, but this can likely be resolved by rearranging the information. For example, I think it would be helpful to show the map in Figure 10 much earlier in the article, perhaps even in the introduction or at the very beginning of the results section to give context to the region of sampling and rationale for choosing that area and those specific locations, as well as introducing what the abbreviations are that are present in all of the plots. 

Figures: There are some inconsistencies in the figure captions including information about what assay was used for quantitation in each one, for example Figure 1 is missing the assay for how cell counts were determined for P. bahamense specifically. Other figures, like Figure 2A, have annotations that are also not clear (a,b,c for statistical significance). The text in the bar graphs in Figure 6 is also very difficult to read and should be revised to make it more legible. Figure 8 has a typo in panel A with StxA4 (should be sxtA4). stx refers to the molecule, sxt refers to the gene. There are a couple of other instances where this occurs as well in the main text.

References: There seem to be some references missing, for example Page 2 line 46, Page 2 line 79 (sentence about grazing deterrents). 

Minor text edits: Throughout there are some sentences that are in parentheses. I think the parentheses should generally be removed and the sentences integrated into the text. Page 2 sentence 51: "No molecular data exist..." while the previous sentence says that toxin was detected. I would interpret that to mean that there is molecular information (or does this mean genetic information?) Perhaps this sentence could be clarified. There is a spelling error page 6 line 173 (fluctuated), and the species names in section 2.6 on page 8 should be italicized. 

Reviewer 2 Report

Comments and Suggestions for Authors

The research provides a concise overview of the research conducted on Pyrodinium bahamense populations from the Indian River Lagoon, Florida, focusing on toxin dynamics and molecular mechanisms associated with saxitoxin (STX) production. It effectively communicates the goals, methods, and key findings of the study. Here, the author examined the relationship between sxtA4 transcripts and the sxtA4/rbcL transcript ratio, providing insights into toxin monitoring. With revisions for clarity and completeness, it has the potential to make a valuable contribution to the field of harmful algal bloom research. However, publication should be considered after massive corrections.

Major revision

-Please add broader implications of the detecting and monitoring  P. bahamense populations in the Indian River Lagoon. 

-Abstract does not address potential future research directions or recommendations based on the study's outcomes.

-sxtA (italic) represents gene, whilt SxtA means proteins. So please revise all the genes into italic. 

-Figure 9. can't generalize the significant negative correlation was found between stxA4 transcripts and the sxtA4/rbcL ratio. The result only includes the case of DB. 

-Additional correlation analysis of sxtA4 transcripts and the sxtA4/rbcL transcipt ratio in HC, KP, B42. With only data of DB, the title should be changed in to "Toxin Dynamics Among Populations of the Bioluminescent HAB species Pyrodinium bahamense from the Diamond Bay, Indian River Lagoon, FL"

-Line 104. Research result is include in introduction. 

-Please explain why the author analyzed the correlation analysis of sxtA4 transcripts and the sxtA4/rbcL transcipt ratio. And there is no any comparison discussion and/or result between rbcL vs. sxtA4. 

Minor revision 

Several/minor mistakes were found. 

-p value (p) should be written in italic (ex. Line 130)

-cell-1. -1 should be written in upper letter. (ex. line 135)

-"flcutuated" -> "fluctuated") - (Line 172)

-"was was" -> redundant - (Line 136)

-"correlation analaysis" -> should be "analysis" - (Line 267)

-"P. bahamense blooms" - (Line 48)

-"sxtA4 gene not detected via the single-cell multiplex PCRs" - (Line 66)

-abbreviations "DB" and "KP" are used without prior explanation. Also, please use abbreviation in figure, and add details in the legend. 

-per L and L -1. Please unify the units. (line 216)

-Species name shoud be in italic "Amphidinium carterae, Karlodinium veneficum" (Line 228)

-Space. (line 268)

Figures. Add the x- and y-axis information in detail

Comments on the Quality of English Language

There is no comment on English quality, but many mistakes such as scientific grammar, italics, unit expressions, etc. have been found.

Round 2

Reviewer 2 Report

Comments and Suggestions for Authors

Please check overall mauscript for minor revisions mentioned below. 

Line 68. cells, with s -> Please check the underline. 

Line 635. rbcL italic

Line 549.  µL−1 DNA -1 should be written in upper font. Please check all the cases in the manuscript.

Line 532. Provide the primer name of GCCGCCGCCACCATATCC.

Line 416. 10 – 25 L -> 10–25 L

Line 407: P. bahamense  italic (405).  Please check all the cases in the manuscript.

Line 267: 0.967, p = 0.00000020, -> 0.967, p = 0.0000002